# DNASpeech: A Contextualized and Situated Text-to-Speech Dataset with Dialogues, Narratives and Actions

## Abstract

In this paper, we propose contextualized and situated text-to-speech (CS-TTS), a novel TTS task to promote more accurate and customized speech generation using prompts with **D**ialogues, **N**arratives, and **A**ctions (DNA). While prompt-based TTS methods facilitate controllable speech generation, existing TTS datasets lack situated descriptive prompts aligned with speech data. To address this data scarcity, we develop an automatic annotation pipeline enabling multifaceted alignment among speech clips, content text, and their respective descriptions. Based on this pipeline, we present DNASpeech, a novel CS-TTS dataset with high-quality speeches with DNA prompt annotations. DNASpeech contains 2,395 distinct characters, 4,452 scenes, and 22,975 dialogue utterances, along with over 18 hours of high-quality speech recordings. To accommodate more specific task scenarios, we establish a leaderboard featuring two new subtasks for evaluation: CS-TTS with narratives and CS-TTS with dialogues. We also design an intuitive baseline model for comparison with existing state-of-the-art TTS methods on our leaderboard. Experimental results indicate the quality and effectiveness of DNASpeech, validating its potential to drive advancements in the TTS field. [1]

## 1 Introduction

Text-to-speech (TTS) aims to convert input text into human-like speech, attracting significant attention in the audio and speech processing community Shen et al. (2018); Ren et al. (2020); Shen et al. (2023); Ju et al. (2024). Previous studies have shown that incorporating more detailed descriptions of the input text is crucial for improving the accuracy of speech synthesis Guo et al. (2023); Li et al. (2022b); Yang et al. (2024). The speaker's contextual information, such as dialogue history, significantly impacts the generated speech Li et al. (2022a); Guo et al. (2021); Liu et al. (2023). Additionally, situated descriptions are also beneficial to enhance the expressiveness of the speech by providing environmental background Lee et al. (2024). Consequently, we propose a new TTS task termed Contextualized and situated Text-To-Speech (CS-TTS), which considers the impact of contextualized and situated descriptions on speech synthesis. By integrating these detailed descriptions, CS-TTS enables more accurate and expressive speech generation, improving the applicability of TTS systems across diverse scenarios.

Recently, prompt-based TTS methods have gained increasing research interest, providing technical support for customized speech generation Li et al. (2024). While formulating detailed descriptions as prompts can potentially address the CS-TTS task, current datasets lack comprehensive prompts that align with text and speech. Their limitations include: (1) Existing prompts with several key phrases lack sufficient contextual descriptions Kim et al. (2021); Guo et al. (2023); (2) Dialogue-only prompts fail to incorporate multifaceted situated descriptions required for precise speech customization Lee et al. (2023); Li et al. (2022a); (3) Limited speaker characters restrict the exploration of various acoustic characteristics in TTS generation.

These constraints render existing datasets insufficient for CS-TTS research. Therefore, we aim to construct a new CS-TTS dataset incorporating more comprehensive contextualized and situated

---

[1]Dataset will be made public once accepted.

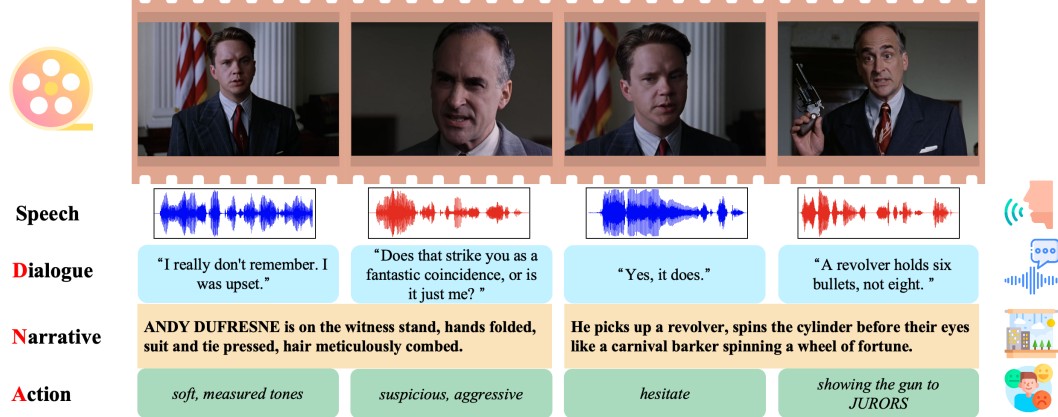

**Speech**

**D**ialogue

**N**arrative

**A**ction

Figure 1: "DNA" descriptions for our proposed CS-TTS task. Dialogues, Narratives, and Actions are annotated to capture the contextualized and situated background essential for TTS generation.

descriptions. As illustrated in Figure 1, we systematically summarize the necessary descriptions into three categories, abbreviated as "**DNA**": (1) **Dialogues** provide the conversational context of speech content; (2) **Narratives** describe the environmental scenes surrounding the speaker's speech; and (3) **Actions** detail the speaker's actions and expressions during speech production.

Among various data sources, movies offer a natural solution due to their rich speech content and diverse character timbres. Movie scripts include not only conversational lines but also environmental scenes that guide the speaker's performance, aligning well with our "DNA" descriptions. Taking advantage of this, we develop an automated annotation pipeline for multifaceted alignment among content text, speech clips, and their corresponding "DNA" descriptions. Based on our efforts in processing movie videos and scripts through this pipeline, we finally collect a new CS-TTS dataset DNASpeech that contains 2,395 distinct characters, 4,452 scenes, and 22,975 dialogue utterances, along with over 18 hours of high-quality speech recordings.

To accommodate more specific task scenarios, we establish a leaderboard featuring two new subtasks: CS-TTS with narratives and CS-TTS with dialogues. Both subtasks are used to evaluate the ability of TTS systems to leverage environmental scenes and dialogue context, along with the speaker's actions, to customize speech. We also introduce an intuitive CS-TTS baseline model for comparison with existing representative TTS methods on our leaderboard. Extensive experimental results validate the effectiveness and quality of DNASpeech, contributing to the advancements of prompt-based TTS.

Our main conclusions can be summarized as follows:

• To support research in CS-TTS, we collect a novel dataset DNASpeech, containing high-quality speech recordings annotated with comprehensive "DNA" prompts: dialogues, narratives, and actions.

• We elaborately present an automatic annotation pipeline for multifaceted alignment among content text, speech clips, and their corresponding descriptions, enabling the efficient collection of high-quality aligned TTS data.

• We establish a leaderboard featuring two new subtasks: CS-TTS with narratives and CS-TTS with dialogues. We also propose an intuitive baseline model for the CS-TTS task. Comprehensive experimental results indicate the quality and effectiveness of DNASpeech.

## 2 RELATED WORK

### 2.1 TEXT-TO-SPEECH WITHOUT PROMPTS

Text-to-speech (TTS) systems have been significantly propelled by the availability of diverse and extensive speech datasets. LJSpeech Ito & Johnson (2017) stands out with its 13,100 high-quality short speech clips of a single speaker, derived from readings of passages from seven non-fiction books.

Another key resource is the LibriSpeech corpus Panayotov et al. (2015), an extensive collection encompassing approximately 1,000 hours of audiobook recordings from the LibriVox project Kearns (2014).

To expand these resources, LibriTTS Zen et al. (2019) offers a multi-speaker English corpus with around 585 hours of read speech, recorded at a 24kHz sampling rate, enhancing the variability and richness of the speech data available for TTS research. The CSTR VCTK Corpus [2] further diversifies the available data with contributions from 110 English speakers exhibiting various accents, each providing approximately 400 sentences sourced from diverse texts, such as newspapers and accent elicitation passages. Moreover, the Hi-Fi Multi-Speaker English TTS Dataset (Hi-Fi TTS) Bakhturina et al. (2021) delivers a robust multi-speaker dataset, consisting of approximately 291.6 hours of speech from 10 speakers, with each contributing at least 17 hours of recordings. These datasets collectively furnish a rich foundation for developing and refining TTS systems, enabling significant improvements in the naturalness and intelligibility of synthetic speech.

## 2.2 TEXT-TO-SPEECH WITH PROMPTS

With the advancement of TTS technology, there has been an increasing emphasis on using prompts to guide speech generation, enabling a more diverse and customized generation process. Initially, seminal works Adigwe et al. (2018); Livingstone & Russo (2018); Zhou et al. (2021) identify the presence of emotional information in speech and construct corresponding datasets by annotating speech with emotions. However, these datasets primarily focus on emotional labels within speech and categorize them into a limited number of classes. To achieve more comprehensive representations, FSNR0 Kim et al. (2021) introduces 327 different labels covering a variety of emotions, intentions, tones, and speech rates. To further advance prompt-based TTS, the PromptSpeech dataset from PromptTTS Guo et al. (2023) utilizes continuous text to describe speech across multiple dimensions, including gender, pitch, loudness, speech rate, and emotion. Similarly, NLSpeech Yang et al. (2024) and TextrolSpeech Ji et al. (2024) employ continuous text descriptions of speech, incorporating more detailed and daily expressions.

The datasets mentioned above mainly focus on describing the speech, lacking contextual information crucial for speech generation. Despite these advancements, datasets with contextual prompts remain relatively scarce. DailyTalk Lee et al. (2023) is a highly popular dataset consisting of 20 hours of speech data from 2,541 dialogues, spoken by two fluent English speakers, a male and a female. The dialogues in DailyTalk are sampled from another dialogue dataset DailyDialog Li et al. (2017). ECC Li et al. (2022a) collects 24 hours of speeches from 66 conversational videos from YouTube. Each dialogue has a duration of 79.3 seconds and features around 2.9 speakers on average. In contrast, MM-TTS Li et al. (2024) highlights the influence of environmental information on speech, amassing expressive speech from film and television data, aligned with corresponding facial expressions and actions.

As shown in Table 1, unlike existing contextual prompt-based TTS datasets, our DNASpeech systematically integrates and aligns three distinct types of descriptive prompts, providing more comprehensive contextualized and situated information to enhance the richness and relevance of the generated speech. Moreover, DNASpeech presents a substantial enhancement in speaker diversity, enabling the exploration of various acoustic characteristics in TTS generation.

Table 1: Comparisons between DNASpeech and existing contextual prompt-based TTS datasets.

| Dataset | Dialogues | Narratives | Actions | Open-Source | #Speakers | #Hours |
|---|---|---|---|---|---|---|
| DailyTalk Lee et al. (2023) | ✗ | ✓ | ✗ | Yes | 2 | 21.67 |
| ECC Li et al. (2022a) | ✗ | ✓ | ✗ | Yes | 673 | 21.12 |
| MM-TTS Li et al. (2024) | ✗ | ✗ | ✓ | No | - | - |
| **DNASpeech (Ours)** | ✓ | ✓ | ✓ | **Yes** | **2395** | 18.37 |

---

[2]https://datashare.ed.ac.uk/handle/10283/3443

## 3 DATASET DESCRIPTION

### 3.1 OVERVIEW

**What is DNASpeech?** We aim to construct a pioneering prompt-based TTS dataset tailored for the CS-TTS task. The proposed dataset DNASpeech aggregates a significant corpus of speech clips sourced from movies and their accompanying scripts. Each speech clip is aligned with three types of prompts: dialogues (D), narratives (N), and actions (A). These prompts, collectively referred to as "DNA", are intricately intertwined with the corresponding speeches, enhancing the contextual richness and situational relevance of the dataset. Specifically, dialogues contain the conversational context preceding the speech; narratives depict the environmental scenes surrounding the speech; and actions describe the speaker's actions and expressions during speech production.

**Why are contextualized and situational prompts necessary?** Textual prompts serve as crucial directives for controlling speech generation, guiding the extraction of emotional and acoustic features necessary for speech synthesis. However, current datasets typically employ direct prompts, which explicitly describe the desired speech attributes such as "Angry, High pitch, Low speed, Loudly." These prompts essentially function as speech annotations and may not always be readily available, particularly in scenarios like audiobooks where detailed prompts are lacking Anguera et al. (2011). In contrast, contextual prompts are closely associated with speech and reflect the situational context in which the speech occurs. For instance, the speech in a spooky and fearful scene is expected to convey low-pitched and tense tones. Despite their prevalence, datasets incorporating such contextualized and situated prompts remain scarce in the field of TTS. Moreover, contextualized prompts require TTS systems to identify subtle nuances of the surrounding context. Therefore, the inclusion of contextual prompts holds promise for driving advancements in TTS technology by enabling more contextually appropriate and natural speech synthesis.

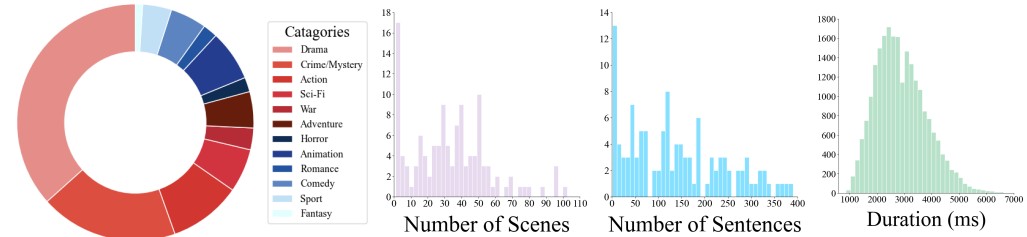

Figure 2: **The DNASpeech Dataset.** *Pie Chart:* Proportion of movie categories. *Histograms, from left to right:* Distribution of the number of scenes, sentences, and speech clip duration in movies. Best viewed online and zoomed in.

### 3.2 DATASET CONSTRUCTION PIPELINE

To efficiently and automatically annotate descriptive prompts aligned with text and speech, we develop a new annotation pipeline. Fig 3 illustrates the overview of this pipeline for DNASpeech, which consists of five fundamental steps: (1) data collection, (2) information extraction, (3) cross-modal alignment, (4) speech denoising, and (5) automatic speech recognition. Data collection and information extraction provide and preprocess the raw movie materials. Cross-modal alignment integrates speech and textual descriptions through both coarse-grained and fine-grained alignment processes. Speech denoising and automatic speech recognition ensure the quality of the speeches.

**Step 1: data collection**

Movies serve as an invaluable resource for TTS research due to their rich speech data and detailed contextual information found in corresponding scripts, such as dialogue lines, narrative scenes, and action depictions. Therefore, we choose movies as the primary data source to construct DNASpeech.

Figure 3: The automatic annotation pipeline for DNASpeech consists of five fundamental steps: (1) data collection of movie materials, (2) information extraction of textual content, (3) cross-modal alignment among "DNA" prompts, text, and speech, (4) speech denoising to reduce background noises and (5) automatic speech recognition to ensure the speech quality. An illustrative example from DNASpeech is provided on the right side.

Inspired by the Condensed Movies Dataset (CMD) Bain et al. (2020) compiling a substantial collection of licensed movie clips from the MovieClip YouTube channel [3], we augment our dataset by collecting newly uploaded movies from the MovieClip channel and purchasing additional movies from legitimate sources. Eventually, we collect a total of 126 movies released between 1940 and 2023, spanning up to 14 common movie categories, to enrich the diversity of our dataset.

**Step 2: information extraction** Following collecting the raw movie videos, the next step is to extract the necessary information, including the speaker's voice and its corresponding lines. Subtitles in SRT format [4] contain the content text along with timestamps for the start and end of each speech segment. We leverage timestamps to obtain aligned text-speech pairs. For other subtitles in image format, we employ SubtitleEdit[5], a widely used software to convert image subtitles into text format using Optical Character Recognition (OCR) technology. Once all subtitles are converted into SRT format, we extract the corresponding speech clips from the movie soundtracks, sampled at a rate of 16,000 Hz, thus obtaining both the speech clips and their associated content text.

Next, our focus shifts to movie scripts obtained from the Internet Movie Script Database (IMSDb)[6], a comprehensive repository of thousands of movie scripts. However, original movie scripts are lengthy and unstructured, necessitating parsing into structured units. Following the script writing paradigm, we extract four key elements from each movie script: *Dialogues Narratives*, *Actions*, and *Characters*. Dialogues denote the speaker's conversational context and line content of their speech within a scene. Narratives represent the basic units defining the overall setting of a shot in the movie. Actions provide supplementary details about characters, describing their actions and expressions. Characters denote the actors for each conversational session. This parsing process allows us to gather the contextualized and situated information of speeches in movies.

**Step 3: cross-modal alignment** Prompt-based TTS tasks necessitate aligning each speech with its corresponding prompts, which is crucial for effective speech synthesis. Leveraging the shared content text between speeches and lines provides a foundation for tackling this alignment challenge. However, while it is theoretically straightforward, aligning speeches with lines directly from the script encounters discrepancies in the content text. To address this issue, we implement a two-stage alignment module combining coarse-grained and fine-grained alignment.

**Coarse-grained alignment.** To match each speech with its corresponding line in the script, more than 800 million potential matches are required, which is computationally intensive and increases the cost of manual verification. Hence, we initially filter out pairs with low textual similarity by performing coarse-grained matching. To be more specific, we preprocess both speech and script

---

[3]https://www.youtube.com/c/MOVIECLIPS

[4]https://docs.fileformat.com/video/srt/

[5]https://www.nikse.dk/subtitleedit

[6]https://imsdb.com/

content by removing stop words, punctuation, and lemmatizing words. We then employ the Longest Common Subsequence (LCS) method to compute textual similarity, retaining *(speech, text)* pairs with a similarity score of 0.9 or higher for subsequent fine-grained alignment.

**Fine-grained alignment.** After coarse-grained alignment, we obtain approximately 30,000 *(speech, text)* pairs. However, the overlap between textual strings may not adequately capture the alignment degree between speech and text. Therefore, in this stage, we utilize the official sentence model `all-mpnet-base-v2`[7] presented by sentence-transformers group to calculate the semantic similarity between speech and text. Pairs with a semantic similarity score of 0.7 or higher are retained. Finally, this process yields 22,975 *(speech, text)* pairs, totaling 18.37 hours of speech data.

**Step 4: speech denoising** The speech clips extracted from the movies in Step 2 usually contain background noises that degrade the quality of the human voice. Therefore, it is essential to separate the human voice from the background noise. Additionally, the speech may sometimes be unclear due to the filming environment, which makes it also important to further enhance the human voice. To eliminate these disturbing noises, we employed Resemble Enhance[8], a common tool designed for noise reduction and speech enhancement. This tool comprises a denoiser and an enhancer, which extract human voices from complex background noise and further improve perceived audio quality by restoring audio distortions and extending the audio bandwidth. Both models are trained using high-quality 44.1kHz voice data, ensuring superior speech enhancement.

**Step 5: automatic speech recognition**

Although speech clips are extracted from movies based on their corresponding subtitle timestamps, discrepancies in duration and clarity may arise, especially in complex dialogue scenes and extended speeches. In addition, denoising speeches can sometimes distort human voices, making them challenging to recognize amidst background noise. To ensure the quality and accuracy of the extracted speeches, it is necsssary to verify them against two criteria: (1) their recognizability and (2) alignment between their content text and the corresponding subtitles. We employ Automatic Speech Recognition (ASR) technology and make the reasonable assumption that if a speech clip can be accurately transcribed by an ASR model, it can also be recognized by humans. We use OpenAI's whisper-large-v3[9] for automatic speech recognition. Samples that do not match their corresponding subtitles after the ASR transcription are eliminated. With this validation process, we finish the construction pipeline of DNASpeech, ensuring its integrity and reliability for subsequent research.

### 3.3 MANUAL ASSESSMENT

After a series of rigorous filtering and screening processes in the pipeline, the quality of samples in DNASpeech generally meets our requirements. Next, further manual assessment is implemented to ensure the high quality of the data and consistency in the subjective evaluation of multiple evaluators. We manually evaluate each sample and assign scores ranging from 1 to 3 based on the overall quality of the sample. The specific criteria for scoring include (1) clarity; (2) emotional richness; (3) speech speed, avoiding excessively fast or slow pacing and (4) the relevance of the speech to the contextual information. Evaluators first score the samples based on each criterion independently, disregarding the other factors. Subsequently, we aggregate the evaluators' scores to obtain an overall quality assessment of each sample and the mean evaluation score for DNASpeech is 2.02.

### 3.4 STATISTICS

We analyze the statistics of speeches, focusing on both pitch and speed to overall present DNASpeech. We extract the F0 fundamental frequency from speeches to obtain their pitch. As shown in Fig 4, the pitch distribution range for female speakers is wider than that for male speakers, evenly distributed from 70Hz to 150Hz; in contrast, the pitch for male speakers is more concentrated, mostly appearing in the 65Hz-95Hz range. Overall, the pitch of female speakers is generally higher than that of male speakers. To more accurately measure the speed of a speech, we calculate the syllables per second (SPS) after removing its silent segments. The distribution shown in the figure indicates that the speakers' speech speed ranges from 6 SPS to 22 SPS, with the 12-15 SPS being the most frequent.

---

[7]https://huggingface.co/sentence-transformers/all-mpnet-base-v2

[8]https://github.com/resemble-ai/resemble-enhance

[9]https://huggingface.co/openai/whisper-large-v3

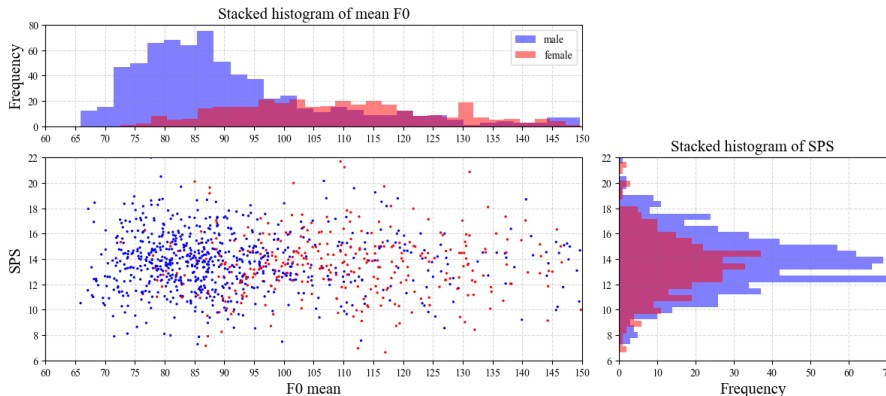

Figure 4: The statistical distribution of the mean F0 and SPS. Each point in the scatter figure represents a speaker. The top and right figures are stacked histograms of mean F0 and SPS by gender.

## 4 EXPERIMENT

### 4.1 COMPARISON METHODS

#### 4.1.1 EXISTING BASELINES.

To evaluate the CS-TTS task, we select several representative text-to-speech methods as baselines for comparison. Based on the input data format and the architecture of models, we categorize these baselines into 3 types: **(1) None-Prompt TTS**, including Tacotron2 Shen et al. (2018), FastSpeech2 Ren et al. (2020), StyleTTS Li et al. (2022b), StyleSpeech Min et al. (2021). **(2) Prompt based TTS**, including PromptTTS2 Leng et al. (2023), PromptTTS++ Shimizu et al. (2024), InstructTTS Yang et al. (2024), VoiceLDM Lee et al. (2024). **(3) Codec model based TTS**, including VALL-E Wang et al. (2023), NaturalSpeech2 Shen et al. (2023), VoiceCraft Peng et al. (2024).

More details about these baselines are introduced in Appendix C and Appendix D.

#### 4.1.2 PROPOSED BASELINE.

Since previous works are not tailored for the CS-TTS task, we design an intuitive baseline model to better evaluate the proposed benchmark. As shown in Fig 5, our baseline model draws from the structure of PromptTTS Li et al. (2022b) and consists of five main modules: Phoneme Encoder, Context Encoder, Style Fusion, Variance Adaptor, and Generator. The Phoneme Encoder uses BERT Devlin et al. (2019) to encode the phonemes of the speech The Context Encoder shares the same structure as the Phoneme Encoder but includes classification tasks for emotion, pitch, energy, and speed during training. To ensure that the generated speech accurately reflects the contextualized and situated descriptions provided in the prompts, we introduce a Style Fusion module that employs a cross-attention mechanism for fine-grained feature fusion.

Given that prompts in the CS-TTS task do not include descriptions of acoustic features, we insert a speaker embedding into the fused representation to control the characteristics of the speech. Inspired by the setup of FastSpeech2 Ren et al. (2020), we incorporate a Variance Adaptor module following the Style Fusion. This module predicts information such as duration, pitch, and loudness, further clarifying the speech characteristics and addressing the one-to-many problem in prompt-based TTS tasks. The final output of our baseline model is a mel-spectrogram, which is transformed into speech using a pre-trained HiFiGAN Kong et al. (2020), ensuring high-fidelity speech synthesis.

### 4.2 DATA QUALITY VERIFICATION

Although the primary purpose of DNASpeech is to aid in CS-TTS task, its inherent text-to-speech mappings make it also suitable for general TTS tasks. Therefore, we can verify its quality by examining the performance of DNASpeech on general TTS tasks. To demonstrate this, we select two

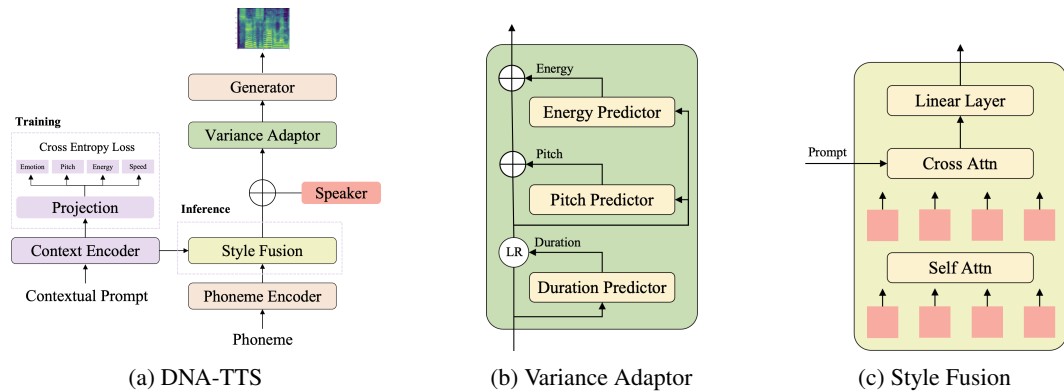

(a) DNA-TTS      (b) Variance Adaptor      (c) Style Fusion

Figure 5: Illustration of the architecture of the proposed baseline for CS-TTS tasks.

TTS models: Tacotron2 and FastSpeech2, along with our baseline model DNA-TTS. Besides, we choose LJSpeech Ito & Johnson (2017) and DailyTalk Lee et al. (2023) as the comparison datasets. For DNASpeech, we first clustered the data by speaker, then randomly sampled 90% of the examples from each speaker for the training set, with the remaining 10% forming the test set. By comparing the performance of these models on DNASpeech with their performance on the comparison datasets, we can assess the effectiveness of DNASpeech as a general TTS dataset.

Following the same setting as DailyTalk, we use mean opinion score (MOS) test as our evaluation metrics. MOS requires evaluators to rate the overall quality of the speech from 1 to 5, with higher scores representing better quality. Three listeners participated in the evaluation process, each holding a master's degree and having completed prior training. After each round of testing, we calculate the Kendall's W coefficient for the scores provided by the three listeners. The results are accepted only when the Kendall's W coefficient $\geq 0.5$, ensuring consistency in the ratings. Results in Table 2 show that models trained on DNASpeech sound as natural as those trained on other datasets, which proves the data quality of DNASpeech.

Table 2: TTS integrity test result for DNASpeech. Score from 1 to 5. A higher score indicates better speech quality. GT refers to the speeches converted from ground truth mel-spectrograms.

| Model | LJSpeech | DailyTalk | DNASpeech |
|---|---|---|---|
| GT | $4.07 \pm 0.08$ | $3.97 \pm 0.07$ | $4.05 \pm 0.08$ |
| Tacotron2 | $3.87 \pm 0.09$ | $3.85 \pm 0.10$ | $3.90 \pm 0.07$ |
| FastSpeech2 | $3.98 \pm 0.07$ | $3.97 \pm 0.08$ | $4.01 \pm 0.07$ |

### 4.3 LEADERBOARD

#### 4.3.1 CS-TTS WITH NARRATIVES

Previous work has been limited by the form of prompts, typically only considering prompts that directly describe speech and lacking the ability to utilize environment information Guo et al. (2023); Leng et al. (2023); Yang et al. (2024). Therefore, we propose CS-TTS with narratives as our first benchmark. We maintain the same training and testing sets as mentioned in Chapter 4.2. For each sample, its environment description is adopted as the input prompt.

To better assess speech quality, our MOS evaluations focus on different aspects: MOS-E emphasizes the alignment of the speech with the environment description, including volume, timbre, and conveyed emotion, aiming to test the ability to utilize information within the environment description. MOS-C focuses on the consistency of the speech itself, with the goal of evaluating the stability of the model when generating speech with the environment description.

The evaluation results are presented in Table 3. We find that: (1) Compared to none-prompt TTS methods, prompt-based methods perform better on the MOS-E metric. We believe this is because

these methods can incorporate additional information from the environment descriptions. (2) For prompt-based methods, MOS-E and MOS-C metrics are generally correlated, indicating that models with a strong ability to capture information in environment description tend to also adhere more closely to its control.

### 4.3.2 CS-TTS with dialogues

Although previous work has explored the use of dialogue to control speech generation Li et al. (2022a); Guo et al. (2021); Liu et al. (2023), they primarily focus on the content of the dialogue itself, neglecting the influence of the conversational scenario (e.g., the speaker's actions and expressions). Therefore, we propose CS-TTS with dialogues, which utilizes the speaker's action states as supplementary information to simulate the scenario of live conversations.

We first use MOS-D to assess the coherence between the speech and the dialogue context. During the evaluation, we primarily consider two factors: the overall emotional tone of the dialogue and the content of the most recent dialogue turn. To evaluate the impact of the action states on the speech, we employ MOS-S to determine whether the speech aligns with the action states. In this assessment, evaluators are initially provided with the dialogue context and action states to infer the speech's emotion, pitch, volume, etc., before listening to the generated speech. They then evaluate the degree of alignment between the two and provide a final score.

From the experimental results presented in Table 3, we can observe the following: (1) Prompt-based methods perform better in terms of MOS-D, indicating that the dialogue context is beneficial for simulating speech expression. (2) There is no significant correlation between performance on MOS-S and MOS-D, which may be attributed to the complexity of conversational scenarios.

Table 3: Leaderboard results of DNASpeech. MOS-E and MOS-C are metrics of CS-TTS with narratives. MOS-D and MOS-S are metrics of CS-TTS with dialogues.

| Model | MOS-E | MOS-C | MOS-D | MOS-S |
|---|---|---|---|---|
| GT | $4.19 \pm 0.07$ | $4.23 \pm 0.08$ | $4.03 \pm 0.08$ | $3.97 \pm 0.10$ |
| Tacotron2 | $3.86 \pm 0.05$ | $3.92 \pm 0.09$ | $3.73 \pm 0.06$ | $3.65 \pm 0.07$ |
| FastSpeech2 | $3.84 \pm 0.08$ | $3.97 \pm 0.13$ | $3.75 \pm 0.09$ | $3.69 \pm 0.09$ |
| StyleTTS | $3.92 \pm 0.11$ | $3.93 \pm 0.07$ | $3.78 \pm 0.07$ | $3.72 \pm 0.06$ |
| StyleSpeech | $3.89 \pm 0.08$ | $3.90 \pm 0.09$ | $3.77 \pm 0.09$ | $3.72 \pm 0.11$ |
| PromptTTS2 | $3.93 \pm 0.07$ | $3.92 \pm 0.11$ | $3.83 \pm 0.11$ | $3.80 \pm 0.07$ |
| PromptTTS++ | $3.93 \pm 0.09$ | $3.99 \pm 0.10$ | $3.78 \pm 0.08$ | $3.70 \pm 0.09$ |
| InstructTTS | $3.94 \pm 0.09$ | $4.12 \pm 0.08$ | $3.83 \pm 0.13$ | $3.75 \pm 0.08$ |
| VoiceLDM | $3.94 \pm 0.07$ | $3.86 \pm 0.06$ | $3.83 \pm 0.09$ | $3.72 \pm 0.08$ |
| VALL-E | $3.89 \pm 0.06$ | $3.95 \pm 0.09$ | $3.76 \pm 0.05$ | $3.74 \pm 0.09$ |
| NaturalSpeech2 | $3.92 \pm 0.04$ | $4.03 \pm 0.07$ | $3.82 \pm 0.05$ | $3.79 \pm 0.06$ |
| VoiceCraft | $3.94 \pm 0.08$ | $4.16 \pm 0.10$ | $3.88 \pm 0.06$ | $3.89 \pm 0.07$ |
| DNA-TTS (Ours) | $3.96 \pm 0.09$ | $4.01 \pm 0.13$ | $3.85 \pm 0.06$ | $3.83 \pm 0.07$ |

## 5 Discussion

In this work, we introduce Contextualized and Situated Text-to-Speech (CS-TTS), aiming to generate speech that adapts to its surrounding context. To address the limitations of existing datasets, which do not sufficiently support CS-TTS research, we collected a new dataset called DNASpeech to facilitate the development of CS-TTS. This dataset contains high-quality speech recordings annotated with "DNA" contextualized and situated prompts: dialogues, narratives, and actions.

Furthermore, we establish a leaderboard to compare the performance of various TTS models on the CS-TTS task. Since there is currently a lack of models specifically designed for CS-TTS, we propose a baseline method to serve as a reference for future research in this area. The results indicate that incorporating contextual information can further enhance the performance of TTS models, with more advanced models showing greater improvements. We believe that our dataset can drive progress in TTS research, moving toward generating smooth and natural speech without manual intervention.

ETHICS STATEMENT

We confirm that we adhere to the ICLR Code of Ethics as stated here. We have taken ethical considerations into account at various stages of our work. The licenses for the datasets contributed in this work are discussed in Appendix A.

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

## A  LICENSE

The dataset is available for free download and non-commercial use under the CC BY-NC-SA 4.0 license.

## B  LIMITATIONS, FUTURE WORK AND SOCIAL IMPACT

**Limitations and Future Work**    There are two main key aspects we aim to address in our future work.  Firstly, DNASpeech collects speech data from movie scenes rather than from real-world scenarios, which might affect the characteristics of the speech. We plan to diversify our dataset by incorporating speech data from more varied and real-world contexts to better reflect authentic speech patterns. Additionally, although we define more comprehensive contextualized and situated prompts than previous TTS datasets, it does not cover all possible prompt types. We intend to explore and integrate additional types of textual prompts to further enrich the dataset, enhancing its utility for a wider range of TTS applications.

**Social Impact**    Given the sensitive nature of biometric data, particularly vocal recordings, all data undergo anonymization to protect personal privacy. However, despite these measures, there exists a potential risk of misuse. To prevent unauthorized usage or dissemination, access to the dataset is subject to a rigorous review process. Regarding the intended use, users are permitted to define their own tasks in our dataset under the license, upon advanced contact with us.

## C  BASELINE DETAILS

**Tacotron2** Shen et al. (2018) is a representative autoregressive TTS models, which composed of a recurrent sequence-to-sequence feature prediction network that maps character embeddings to mel-scale spectrograms.

**FastSpeech2** Ren et al. (2020) is a non-autoregressive TTS model that introduce more variation information (e.g. pitch and energy) of speech and better solves the one-to-many mapping problem in TTS.

**PromptTTS2** Leng et al. (2023) utilizes prompts to guide the speech generation process. It incorporates a variation network that supplies information about voice variability that not captured by the content text.

**PromptTTS++** Shimizu et al. (2024) is designed to synthesize the acoustic characteristics of various speakers based on natural language descriptions. This method employs an additional speaker prompt to efficiently map natural language descriptions to the acoustic features of different speakers.

**InstructTTS** Yang et al. (2024) uses natural language as style prompt to control the styles in the synthetic speech.  It models acoustic features in discrete latent space and train a novel discrete diffusion probabilistic model to generate vector-quantized (VQ) acoustic tokens rather than the commonly-used mel spectrogram.

**StyleSpeech** Min et al. (2021) propose a self-supervised style enhancing method with VQ-VAE-based pre-training for expressive audiobook speech synthesis.

**StyleTTS** Li et al. (2022b) is a generative model designed for parallel text-to-speech (TTS) synthesis, which incorporates innovative techniques, including the Transferable Monotonic Aligner (TMA) and duration-invariant data augmentation methods.

**VoiceLDM** Lee et al. (2024) is designed to produce speech that accurately follows the overall environmental context of the audio. Based on latent diffusion models, it can incorporate an additional descriptive prompt as a conditional input.

**VALL-E** Wang et al. (2023) train a neural codec language model using discrete codes derived from an off-the-shelf neural audio codec model, and regard TTS as a conditional language modeling task

**NaturalSpeech2** Shen et al. (2023) is a TTS system that leverages a neural audio codec with residual vector quantizers to get the quantized latent vectors and uses a diffusion model to generate these latent vectors conditioned on text input.

**VoiceCraft** Peng et al. (2024) employs a Transformer decoder architecture and introduces a token rearrangement procedure that combines causal masking and delayed stacking to enable generation within an existing sequence.

## D  TRAINING PARAMETERS

| Model | Optimizer | $\beta_1$ | $\beta_2$ | $\epsilon$ | Batch size | Training steps | Learning rate |
|-------|-----------|-----------|-----------|------------|------------|----------------|---------------|
| Tacotron2 | Adam | 0.9 | 0.99 | $10^{-6}$ | 16 | 2 epochs | $10^{-4}$ |
| FastSpeech2 | Adam | 0.9 | 0.98 | $10^{-9}$ | 16 | 2 epochs | $10^{-5}$ |
| StyleTTS | AdamW | 0 | 0.99 | $10^{-7}$ | 16 | 2 epochs | $10^{-4}$ |
| StyleSpeech | Adam | 0.9 | 0.98 | $10^{-9}$ | 16 | 2 epochs | $2 \times 10^{-4}$ |
| PromptTTS2 | Adam | 0.9 | 0.99 | $10^{-7}$ | 16 | 2 epochs | $10^{-5}$ |
| PromptTTS++ | Adam | 0.9 | 0.99 | $10^{-7}$ | 16 | 2 epochs | $10^{-5}$ |
| InstructTTS | AdamW | 0.9 | 0.94 | $10^{-7}$ | 16 | 2 epochs | $3 \times 10^{-6}$ |
| VoiceLDM | AdamW | 0.9 | 0.99 | $10^{-7}$ | 16 | 2 epochs | $2 \times 10^{-5}$ |

Table 4: Training configurations for different models

| Model | Schedule | Other params |
|-------|----------|--------------|
| Tacotron2 | / | / |
| FastSpeech2 | Linear schedule | Warm up step=200 |
| StyleTTS | OneCycleLR | Weight decay=$10^{-4}$, $\lambda_{s2s} = 0.2$, $\lambda_{adv} = 1$, $\lambda_{mono} = 5$, $\lambda_{fm} = 0.2$, $\lambda_{dur} = 1$, $\lambda_{f0} = 0.1$, $\lambda_n = 1$ |
| StyleSpeech | / | / |
| PromptTTS2 | / | / |
| PromptTTS++ | / | / |
| InstructTTS | Linear schedule | Warm up step=200 |
| VoiceLDM | / | Drop rate of $c_{desc}$=0.1, Drop rate of $c_{cont}$=0.1 |

Table 5: Training configurations for different models

