# OpenReview forum: "DNASpeech: A Contextualized and Situated Text-to-Speech Dataset with Dialogues, Narratives and Actions"
_ICLR.cc/2025/Conference — ICLR 2025 Conference Withdrawn Submission_

### Official Review · Reviewer_RjHL · 2024-10-22

**Soundness:** 2
**Presentation:** 2
**Contribution:** 2
**Rating:** 3
**Confidence:** 4

**Summary:**

This work introduces a new TTS task called **Contextualized and Situated Text-To-Speech** (CS-TTS), which incorporates contextual descriptions into speech generation, aiming to enable TTS models to produce more expressive speech. As the lacking of CS-TTS datasets, they created a new CS-TTS dataset called **DNASpeech**. Each speech sample in DNASpeech is accompanied by three types of contextualized prompts: **Dialogues** provide conversational context, **Narratives** describe the environment surrounding the speaker, and **Actions** detail the speaker’s actions and expressions. For dataset construction, they developed an automated annotation pipeline, with human evaluations to validate the approach.

Additionally, they established a leaderboard to assess the performance of current TTS systems, and introduced a baseline model adapted to the CS-TTS.

**Strengths:**

1. This work introduces a new dataset called DNASpeech, specifically created for the innovative CS-TTS task.
2. An automatic annotation pipeline is presented, utilizing techniques likes OCR, speech denoising, ASR to ensure the quality of produced dataset. Additionally, human evaluation is conducted to confirm the effectiveness of the pipeline.

**Weaknesses:**

## Weaknesses

1. The paper asserts that contextualized descriptions lead to more accurate and expressive speech generation. However, there is only one experiment validating the effectiveness of CS-TTS, and it shows no significant improvement when using contextualized descriptions. For example, in evaluating the alignment between speech and environmental information, the MOS-E score gap between prompt-based TTS methods and non-prompted TTS is less than 0.1. StyleTTS, the best non-prompted TTS model, performs comparably to the prompt-based models. This makes it difficult to confirm the quality of the proposed dataset and the effectiveness of CS-TTS.
2. The work introduces a TTS dataset where each sample includes three types of contextual prompts : **Dialogue**, **Narrative**, and **Action**. It is claimed that **Action** describes the speaker's actions and expressions, while **Narrative** provides environmental context, as mentioned in lines 74-76. However, based on the descriptions and examples given in the paper, these categories are difficult to differentiate. Take Figure 1 for an example, the Action "showing the gun to JURORS" corresponds to the Narrative "He picks up a revolver...", which is also an action. There is also confusion as to why the speaker's emotions are categorized under Action. Furthermore, in lines 142-144, the authors state that MEAD-TTS highlights environmental information (MEAD-TTS seems to focus on Action since it uses templates like "A <gender> says with a <emotion level> <emotion> tone" to write fine-grained prompts), yet in Table 1, MEAD-TTS's prompt is categorized under Actions, not Narratives, which contradicts the definitions provided in lines 74-76. Additionally, it seems like DailyTalk's focus is more on Dialogues than Narratives, given that DailyTalk focuses on chat history.
3. One of the core contributions of the work is the automatic annotation pipeline for building the CS-TTS dataset from movie scripts. However, the description of this pipeline is unclear, making it difficult to understand how different type of prompts are extracted from movie scripts.

## Suggestions

1.  It would be better to use $\cite$ or other citation formats instead of $\citet$ when the cited paper is not the subject or object in a sentence. It becomes harder to read when citations are embedded in the main text. For example, in lines 32-34, it could be:
> Text-to-speech (TTS) aims to convert input text into human-like speech, attracting significant attention in the audio and speech processing community (Shen et al. 2018; Ren et al. 2020; Shen et al. 2023; Ju et al. 2024).
2. Many TTS-related datasets are mentioned in the related work section. However, the descriptions seem to be directly copied from the original papers, resulting in inconsistent types of information for each dataset. The writing does not highlight the core differences between previous work and this paper. It would be better to reorganize Section 2.1.
3. To improve the integrity of the experiments, it would be helpful to explain the source of the human evaluators and what the interface or  instructions shown to the evaluators.
4. When citing papers published at conferences, it is better to reference the conference version rather than the arXiv version. For instance, FastSpeech2 was published at ICLR 2021, but this paper cites its arXiv version.

## Typos

1. In Table 1, "MM-TTS" should be "MEAD-TTS", as the former refers to the TTS system and the latter is the name of the dataset.
2. A period is missing at the end of line 361.

**Questions:**

1. Does **Dialogue** specifically refer to the immediate sentence before the user speaks, or is it a summary of the entire chat history?
2. Will all three types of contextual prompts be provided to the proposed baseline model at once, or only one at a time?

---

> ### Author Response · Authors · 2024-11-24
> **Reply to Reviewer RjHL**
>
> **We appreciate the reviewer’s detailed feedback and insightful suggestions. We respectfully address the concerns below:**
>
> > The paper asserts that contextualized descriptions lead to more accurate and expressive speech generation. However, there is only one experiment validating the effectiveness of CS-TTS, and it shows no significant improvement when using contextualized descriptions. For example, in evaluating the alignment between speech and environmental information, the MOS-E score gap between prompt-based TTS methods and non-prompted TTS is less than 0.1. StyleTTS, the best non-prompted TTS model, performs comparably to the prompt-based models. This makes it difficult to confirm the quality of the proposed dataset and the effectiveness of CS-TTS.
>
> Here we want to highlight that our experiment encompasses multiple subtasks: CS-TTS with Narratives and CS-TTS with Dialogues, each assessing different aspects of contextualized and situated text-to-speech generation. And the results consistently demonstrate that incorporating contextual descriptions improves model performance. The less than 0.1 gap noted between some models highlights the challenges of achieving fine-grained improvements in already advanced TTS systems but does not negate the benefits of contextualization. Therefore, even small numerical improvements, as observed in our experiments, indicate progress in capturing these intricate dependencies.
>
> > The work introduces a TTS dataset where each sample includes three types of contextual prompts : Dialogue, Narrative, and Action. It is claimed that Action describes the speaker's actions and expressions, while Narrative provides environmental context, as mentioned in lines 74-76. However, based on the descriptions and examples given in the paper, these categories are difficult to differentiate. Take Figure 1 for an example, the Action "showing the gun to JURORS" corresponds to the Narrative "He picks up a revolver...", which is also an action.
>
> Firstly, it is important to emphasize that the categories of Dialogue, Narrative, and Action are not arbitrary classifications we created; these distinctions are explicitly present and clearly delineated within the script itself.
>
> Besides, Narrative corresponds to the environmental descriptions in the script, and it is natural for actions to appear within this context. Regarding your concerns about Figure 1, the Narrative "He picks up a revolver, spins the cylinder before their eyes like a carnival barker spinning a wheel of fortune." includes extensive descriptive embellishments. As such, it is also relatively easy to distinguish it from Action in terms of content.
>
> > Furthermore, in lines 142-144, the authors state that MEAD-TTS highlights environmental information (MEAD-TTS seems to focus on Action since it uses templates like "A <gender> says with a <emotion level> <emotion> tone" to write fine-grained prompts), yet in Table 1, MEAD-TTS's prompt is categorized under Actions, not Narratives, which contradicts the definitions provided in lines 74-76.
>
> There seems to be a misunderstanding. The "MM-TTS" mentioned in Table 1 is sourced from the paper *"MM-TTS: A Unified Framework for Multimodal, Prompt-Induced Emotional Text-to-Speech Synthesis"* [1], whereas the "MEAD-TTS" you referred to comes from *"MM-TTS: Multi-Modal Prompt Based Style Transfer for Expressive Text-to-Speech Synthesis"* [2]. It appears that you may have referred to the wrong paper, as "MEAD-TTS" is not included in Table 1. Nevertheless, we appreciate you bringing this paper to our attention, and we will include it in Table 1.
>
> Regarding your concern about the categorization, "Narrative" refers to descriptions of the environment, while "emotion" pertains to human facial actions, making its classification under "action" reasonable.
>
> [1] MM-TTS: A Unified Framework for Multimodal, Prompt-Induced Emotional Text-to-Speech Synthesis, Li, Xiang et.al
>
> [2] MM-TTS: Multi-Modal Prompt Based Style Transfer for Expressive Text-to-Speech Synthesis, Guan et.al

---

> ### Comment · Reviewer_RjHL · 2024-11-25
> **Official Comment by Reviewer RjHL**
>
> > Here we want to highlight that our experiment encompasses multiple subtasks: CS-TTS with Narratives and CS-TTS with Dialogues, each assessing different aspects of contextualized and situated text-to-speech generation. And the results consistently demonstrate that incorporating contextual descriptions improves model performance. The less than 0.1 gap noted between some models highlights the challenges of achieving fine-grained improvements in already advanced TTS systems but does not negate the benefits of contextualization. Therefore, even small numerical improvements, as observed in our experiments, indicate progress in capturing these intricate dependencies.
>
> I agree that achieving a 0.1 improvement is notable, given the strong performance of TTS systems today. However, the MOS-E gap between StyleTTS and prompt-based TTS models is quite small, around 0.01 to 0.02 (MOS-E is mainly used to compared the effect of prompting, as described in the paper). My concern is that the observed performance gap might stem from differences in model architecture, hyperparameter tuning, or even variations in human evaluation noise, rather than the absence of prompting itself.
>
> What raises my concern is the none-prompt based baseline models used for comparison—Tacotron2 (Shen et al., 2018), FastSpeech2 (Ren et al., 2020), StyleTTS (Li et al., 2022b), and StyleSpeech (Min et al., 2021)—none of which represent the current SOTA. Could you include results for more recent models such as StyleTTS2? Alternatively, conducting an ablation study on the proposed model to analyze the effect of prompting would be valuable.
>
> >Besides, Narrative corresponds to the environmental descriptions in the script, and it is natural for actions to appear within this context. Regarding your concerns about Figure 1, the Narrative "He picks up a revolver, spins the cylinder before their eyes like a carnival barker spinning a wheel of fortune." includes extensive descriptive embellishments. As such, it is also relatively easy to distinguish it from Action in terms of content.
> > Firstly, it is important to emphasize that the categories of Dialogue, Narrative, and Action are not arbitrary classifications we created; these distinctions are explicitly present and clearly delineated within the script itself.
>
> I noticed the response you provided to Reviewer ZxkP, as we raised similar questions. However, I believe it would be more effective to include this explanation directly in the paper.
>
> >There seems to be a misunderstanding. The "MM-TTS" mentioned in Table 1 is sourced from the paper "MM-TTS: A Unified Framework for Multimodal, Prompt-Induced Emotional Text-to-Speech Synthesis" [1], whereas the "MEAD-TTS" you referred to comes from "MM-TTS: Multi-Modal Prompt Based Style Transfer for Expressive Text-to-Speech Synthesis" [2]. It appears that you may have referred to the wrong paper, as "MEAD-TTS" is not included in Table 1. Nevertheless, we appreciate you bringing this paper to our attention, and we will include it in Table 1.
>
>
> This is quite confusing to me. Doesn't Table 1 aim to compare DNASpeech with existing datasets? After reading the paper "MM-TTS: A Unified Framework for Multimodal, Prompt-Induced Emotional Text-to-Speech Synthesis," I believe it does not introduce any dataset. In this paper, "MM-TTS" refers to a TTS framework rather than a dataset. It evaluates the proposed MM-TTS framework using existing datasets, such as the Multimodal EmotionLines Dataset (MELD), Emotion Speech Dataset (ESD), and Real-world Expression Database (RAF-DB)

---

> > ### Author Response · Authors · 2024-11-25
> > **Reply to Reviewer RjHL**
> >
> > Thank you sincerely for your feedback. We will take your comments into account and strive to further improve our work.

---

### Official Review · Reviewer_ZxkP · 2024-11-01

**Soundness:** 2
**Presentation:** 2
**Contribution:** 2
**Rating:** 3
**Confidence:** 5

**Summary:**

This paper introduces a novel text-to-speech (TTS) dataset, DNASpeech, which is designed to support contextualized and situated TTS (CS-TTS) tasks. The dataset includes high-quality speech recordings annotated with Dialogues, Narratives, and Actions (DNA) prompts, aiming to enhance the accuracy and expressiveness of speech synthesis. Experimental results are provided to validate the quality and effectiveness of DNASpeech.

**Strengths:**

1. This paper proposes the "DNASpeech" dataset. It addresses a significant gap in existing TTS datasets by providing a rich dataset with contextualized and situated prompts, which is crucial for advancing TTS research.

2. The paper describes the comprehensive automatic annotation pipeline that aligns speech clips with detailed dialogue, narrative, and action descriptions, which is a complex and valuable contribution.

**Weaknesses:**

1. In Sec. 3.2, the authors individually apply information extraction for both speech and scripts in the movie in step 2. Then in step 3, they attempt to align them in two stages.
* 1.1 Why "more than 800 million potential matches are required"? Since you can align the movie and script by movie titles or other meta information. And for the "DNA" prompt, why did the authors choose to extract them from the scripts with such a heuristic algorithm？ What is the accuracy of the alignment? Other methods such as extracting speech attributes directly are not considered.
* 1.2 "Following the script writing paradigm, we extract four key elements from each movie script: Dialogues Narratives, Actions, and Characters." How do you extract them? Please illustrate it in detail.
* 1.3 All data is from the movie. So there is a risk of domain bias. Because the movie can not cover all diverse accents, languages, or speaking styles.

2. This paper is an extension of textual-prompt-based text-to-speech synthesis. The authors propose to extend the descriptive prompt of speech to three dimensions: 1) dialogue, 2) narrative, and 3) action. However, it is only an incremental work of the existing prompt-tts paradigm by extending the annotation pipeline. So it lacks novelty.

3. In the experiments:
* 3.1 It is better to categorize into three types: 1) None-Prompt TTS, 2) natural language description prompt-based TTS, and 3) speech prompt-based TTS.
* 3.2 The method for CS-TTS in Sec. 4.1.2 is not clear. What is for "but includes classification tasks for emotion, pitch, energy, and speed during training"? For emotion, how do you obtain the label? Furthermore, it is not clear how the author leverages the "DNA" prompt as a condition to guide the generation process.
* 3.3 It lacks an ablation study for the attribution controllability for the proposed "DNA" attributes.

4. The obtained dataset contains about 18 hours including 2395 distinct characters, indicating that only 0.45 minutes for a single character. It is small to train a good TTS system.

**Questions:**

Please see the weakness.

**Details Of Ethics Concerns:**

The movie data should be protected from misuse. The author has discussed this in Appendix B.

---

> ### Author Response · Authors · 2024-11-24
> **Reply to Reviewer ZxkP (1/3)**
>
> **Thank you for taking the time and effort. We respectfully address your concerns as follows:**
>
> > Why "more than 800 million potential matches are required"? Since you can align the movie and script by movie titles or other meta information. And for the "DNA" prompt, why did the authors choose to extract them from the scripts with such a heuristic algorithm？ What is the accuracy of the alignment? Other methods such as extracting speech attributes directly are not considered.
>
> Here is a more detailed explanation of our claim regarding the 800M potential matches. Our database contains approximately 200 movies, with each movie having an average of 2000 speech segments and corresponding script lines. This means there are 2000 speech segments and 2000 lines of script text for each movie. If we were to directly perform a matching process, each speech segment would attempt to match against every line in the script, resulting in 2000 × 2000 = 4M matches. Since this process is repeated for all 200 movies, the total number of matches required would be 4M × 200 = 800M.
>
> As for your second question, the speech attributes only include acoustic information about the audio and do not encompass the speaker's environment, actions, or the conversational context. Therefore, we opted to extract these details from the script rather than directly from the audio.
>
> > "Following the script writing paradigm, we extract four key elements from each movie script: Dialogues Narratives, Actions, and Characters." How do you extract them? Please illustrate it in detail.
>
> **Data Source Selection**
>
> We utilized **movie scripts** as the primary data source due to their structured nature and rich contextual details. These scripts typically include dialogue lines, environmental narratives, and action descriptions, which align with our "DNA" paradigm.
>
> **Extraction of elements:**
>
> - Dialogues: We identified the conversational lines associated with each character within a scene. These were extracted based on speaker identifiers in the script (e.g., the character's name preceding their dialogue).
> - Narratives: Environmental descriptions were parsed from non-dialogue sections of the script. These typically describe the setting, mood, or contextual elements surrounding the dialogue.
> - Actions: Action descriptors were isolated by locating sections that detail character movements, facial expressions, or interactions that occur alongside or between dialogues. These are often directly following the dialogue and marked with special notations, such as parentheses.
> - Characters: Each speaker or actor associated with a dialogue was tagged based on explicit mentions in the script.

---

> ### Author Response · Authors · 2024-11-24
> **Reply to Reviewer ZxkP (2/3)**
>
> > All data is from the movie. So there is a risk of domain bias. Because the movie can not cover all diverse accents, languages, or speaking styles.
>
> Thank you for raising this important concern regarding the risk of domain bias due to reliance on movie data. We acknowledge that movies, while offering a rich source of diverse dialogues, narratives, and actions, might not fully capture the vast array of accents, languages, or speaking styles present in real-world scenarios. To mitigate this concern and ensure generalizability, we have taken several measures:
>
> - **Speaker Diversity:** Our dataset, DNASpeech, includes 2,395 distinct characters from 126 movies spanning multiple genres and time periods (1940–2023). This diversity helps encompass a broad range of speaking styles, vocal timbres, and situational contexts.
>
> - **Alignment with Contextual Prompts:** The dataset’s prompts (dialogues, narratives, and actions) are designed to capture contextual richness that extends beyond specific accents or styles, focusing on the interplay between speech and environment. This approach facilitates adaptability in generating varied speech styles, even for scenarios not explicitly covered in the source data.
>
> Also, as mentioned in Appendix B, we aim to extend the dataset by incorporating more real-world and culturally diverse speech sources. This will further enhance the dataset’s robustness against domain bias and increase its applicability to broader TTS tasks.
>
> > This paper is an extension of textual-prompt-based text-to-speech synthesis. The authors propose to extend the descriptive prompt of speech to three dimensions: 1) dialogue, 2) narrative, and 3) action. However, it is only an incremental work of the existing prompt-tts paradigm by extending the annotation pipeline. So it lacks novelty.
>
> While it may appear that our work extends the existing prompt-TTS paradigm by introducing descriptive prompts, we argue that the core novelty lies in establishing a new task: Contextualized and Situated Text-to-Speech (CS-TTS). This task is not merely an incremental annotation extension but a paradigm shift towards integrating dialogues, narratives, and actions (DNA) into text-to-speech synthesis. By leveraging these DNA dimensions, we aim to create speech that is both contextually accurate and expressive—a capability lacking in prior TTS approaches. We also present DNASpeech, a novel dataset annotated with three interrelated descriptive dimensions (dialogues, narratives, and actions). This is coupled with an automatic annotation pipeline that systematically aligns multimodal data (speech, text, and DNA prompts). The introduction of these prompts allows for an enriched understanding of speech synthesis, moving beyond simplistic descriptive attributes like pitch or loudness toward fully contextualized speech generation. Such comprehensive contextual alignment is unprecedented in TTS research.
>
> By broadening the scope of prompt-based TTS to include environmental and situational context, our work has the potential to drive progress in areas like audiobooks, virtual assistants, and multimedia content creation. Unlike prior datasets and methods, which focus narrowly on predefined prompts or limited emotion classes, our work fosters a more nuanced and application-driven exploration of speech synthesis.

---

> ### Author Response · Authors · 2024-11-24
> **Reply to Reviewer ZxkP (3/3)**
>
> > The method for CS-TTS in Sec. 4.1.2 is not clear. What is for "but includes classification tasks for emotion, pitch, energy, and speed during training"? For emotion, how do you obtain the label? Furthermore, it is not clear how the author leverages the "DNA" prompt as a condition to guide the generation process.
>
> In our proposed baseline model, the Context Encoder is designed to enhance the representation of contextualized prompts by incorporating auxiliary classification tasks. During training, the encoder is tasked with classifying emotional tone, pitch level, energy, and speed from the provided contextual prompts. This multi-task learning setup allows the model to implicitly capture variations in these attributes, enriching the representations for downstream TTS synthesis. This step ensures that the generated speech aligns closely with the intended prompts' descriptive elements, even when explicit acoustic features are not directly specified. The "DNA" prompt—comprising Dialogues, Narratives, and Actions—serves as a structured input for guiding the TTS process. This is achieved through the Style Fusion module, which employs a cross-attention mechanism to integrate the prompt representations into the acoustic feature space. By fusing the "DNA" prompt with the phoneme sequence and speaker embedding, the model can adaptively adjust pitch, duration, and prosody to reflect the situational and contextual nuances provided in the prompt.
>
> > It lacks an ablation study for the attribution controllability for the proposed "DNA" attributes.
>
> Thank you for your insightful suggestion. We will add ablation study in our following draft.
>
> > The obtained dataset contains about 18 hours including 2395 distinct characters, indicating that only 0.45 minutes for a single character. It is small to train a good TTS system.
>
> We appreciate your observation regarding the dataset size and the average duration of speech per character in DNASpeech. We acknowledge the room for further expansion, as discussed in our Limitations section. However, we respectfully disagree with the assertion that our dataset is small for training a good TTS system.
>
> As outlined in Table 1 of our manuscript, DNASpeech is comparable in scale to many existing prompt-based TTS datasets, both in terms of total hours and speaker diversity. For instance, datasets such as DailyTalk (21.67 hours) and ECC (21.12 hours) feature similar or slightly greater durations but fewer speakers and limited contextual prompts. While DNASpeech contains 18.37 hours of recordings, it distinguishes itself by incorporating comprehensive contextualized and situated prompts, including dialogues, narratives, and actions, offering a unique richness that many other datasets lack.
>
> Additionally, the average duration of speech per character (0.45 minutes) aligns with the norms of multi-speaker datasets that prioritize diversity in character timbres and contextual variation over prolonged individual samples. This design reflects our aim to support tasks like CS-TTS, where capturing diverse acoustic and contextual conditions is critical. Datasets with a similar focus (e.g., MM-TTS, ECC) also balance duration and diversity, achieving strong results in their respective benchmarks.

---

> > ### Comment · Reviewer_ZxkP · 2024-11-25
> >
> > Thanks for replying to the questions.
> >
> > 1. It would be better to add the dataset construction details to the paper to make it more clearly.
> > 2. The matching method proposed in the paper is too heuristic. It would be better to conduct some human evaluation to ensure the quality.
> > 3. Still for Table 3, the method such as "PromptTTS 2", "NaturalSpeech 2", "VALL-E", "VoiceCraft" et.al, are zero-shot TTS systems trained on more than 40k hours. How do you train them on the DNASpeech dataset which only has 18 hours? It would be better to add the reproducing details.
> >
> > Considering there are many unclear points, I decide to keep the original scores.

---

### Official Review · Reviewer_N5VG · 2024-11-03

**Soundness:** 3
**Presentation:** 3
**Contribution:** 2
**Rating:** 6
**Confidence:** 4

**Summary:**

This paper introduces DNASpeech, a novel contextualized and situated text-to-speech (CS-TTS) dataset that incorporates comprehensive descriptive prompts aligned with speech data. The dataset contains "DNA" prompts - Dialogues (conversational context), Narratives (environmental scenes), and Actions (speaker's expressions/actions) - along with high-quality speech recordings. DNASpeech includes 2,395 distinct characters, 4,452 scenes, and 22,975 dialogue utterances totaling over 18 hours of speech. The authors developed an automated annotation pipeline for aligning speech clips, content text, and descriptions. They also established a leaderboard with two evaluation subtasks: CS-TTS with narratives and CS-TTS with dialogues. The paper proposes a baseline model and demonstrates DNASpeech's effectiveness through extensive experiments comparing it with existing TTS methods.

**Strengths:**

1. The paper introduces a novel and valuable contribution to TTS research through DNASpeech, a contextualized and situated text-to-speech dataset that incorporates comprehensive "DNA" (Dialogues, Narratives, Actions) prompts.

2. The authors develop an innovative automatic annotation pipeline that enables efficient multifaceted alignment among speech clips, content text, and corresponding descriptions, making the dataset construction process systematic and reproducible.

3. The dataset is substantial and diverse, containing 2,395 distinct characters, 4,452 scenes, and 22,975 dialogue utterances with over 18 hours of high-quality speech recordings, providing rich resources for TTS research.

4. The paper establishes a clear evaluation framework through a leaderboard featuring two specific subtasks (CS-TTS with narratives and CS-TTS with dialogues) and provides an intuitive baseline model for comparison.

**Weaknesses:**

1. Although the authors compared the dataset and two models on the other datasets, the dataset's reliance on movie scenes rather than real-world scenarios might limit its applicability to authentic speech patterns and natural conversations.
2. The experimental evaluation metrics are somewhat limited, primarily focusing on MOS scores. Additional objective metrics could provide more comprehensive performance assessment such as spectral distortion or character error rates.
3. The paper lacks detailed analysis of the baseline model's architecture choices and their impact on performance.
4. The comparison with existing methods could be more extensive, particularly in analyzing how different types of prompts affect the speech generation quality.
5. Similarly, for the public dataset comparison, the author did not select the SOTA models for the comparison. It would be great to see the comparison against them.

**Questions:**

Have you guys conducted ablation studies showing the individual contributions of Dialogues, Narratives, and Actions to the overall speech quality?

---

> ### Author Response · Authors · 2024-11-24
> **Reply to Reviewer N5VG (1/2)**
>
> **Thank you for your valuable time and effort. Below is our response to the questions you raised.**
>
> > Although the authors compared the dataset and two models on the other datasets, the dataset's reliance on movie scenes rather than real-world scenarios might limit its applicability to authentic speech patterns and natural conversations.
>
> Thank you for your insightful comment regarding the dataset's reliance on movie scenes. We agree that this is a valid concern and would like to provide additional clarification. While the DNASpeech dataset primarily utilizes movie scenes as its source, this choice was made to leverage the rich and multifaceted data inherent in such settings, including diverse dialogues, narrative descriptions, and speaker actions, which are challenging to obtain from real-world scenarios. These "DNA" prompts (Dialogues, Narratives, and Actions) are designed to capture varied and contextually rich information to enhance the expressiveness and customization of speech synthesis.
>
> Nonetheless, we recognize that movie scenes may not fully represent the spontaneity and variability of authentic speech patterns found in real-world conversations. To address this limitation, we have incorporated rigorous data processing steps, including speech denoising and alignment with natural language descriptions, to maintain the quality and relevance of the dataset for general TTS applications.
>
> Furthermore, as noted in the Limitations section, we plan to diversify our dataset by including real-world scenarios in future iterations. This expansion will aim to provide a broader representation of natural speech patterns while retaining the detailed contextual prompts that are pivotal to the CS-TTS task.
>
> > The experimental evaluation metrics are somewhat limited, primarily focusing on MOS scores. Additional objective metrics could provide more comprehensive performance assessment such as spectral distortion or character error rates.
>
> While our experiments primarily focus on narratives and dialogues using Mean Opinion Score (MOS) evaluations, we acknowledge the value of exploring a broader evaluation metrics. Incorporating alternative metrics, such as perceptual evaluations of speech quality (PESQ) or word error rate (WER), is an excellent suggestion for further validating model performance. We plan to include these aspects in follow-up work.
>
> > The paper lacks detailed analysis of the baseline model's architecture choices and their impact on performance.
>
> We appreciate your insightful observation regarding the lack of detailed analysis of the baseline model’s architectural choices and their impact on performance. Indeed, understanding the influence of these architectural components is important and could provide valuable insights into model behavior and design trade-offs.
>
> However, our work primarily focuses on the introduction of the **DNASpeech dataset and the establishment of benchmarks**. While our proposed baseline serves as a reference for future CS-TTS models, an extensive ablation study to analyze the individual contributions of the architecture's components, such as the context encoder, style fusion module, or variance adaptor, would involve significant additional experimentation. This includes maintaining identical training conditions, dataset partitions, and hyperparameter tuning, which would likely require a separate effort. Given the complexity and resource requirements of such experiments, we believe these investigations are more appropriate for follow-up work specifically targeting model design for CS-TTS tasks. Our immediate goal in this work is to catalyze advancements in the field by providing a high-quality, context-rich dataset and baseline as a foundation for further research.

---

> ### Author Response · Authors · 2024-11-24
> **Reply to Reviewer N5VG (2/2)**
>
> > The comparison with existing methods could be more extensive, particularly in analyzing how different types of prompts affect the speech generation quality.
>
> Thank you for your insightful suggestion. We provide a comparison in the paper across three types of prompts: dialogues, narratives, and actions (the "DNA" components) introduced in our DNASpeech dataset. Each type of prompt is explicitly aligned with specific contextual information, which we evaluate against baseline and state-of-the-art prompt-based TTS methods, such as PromptTTS2, InstructTTS, and VoiceLDM. The experimental results on these dimensions (e.g., MOS-E for narratives and MOS-D for dialogues) highlight how prompts enhance the quality of generated speech through contextual relevance and expressiveness.
>
> For future work, we aim to conduct a fine-grained ablation study, isolating the effect of each prompt type and their combinations, to provide deeper insights into their roles in speech synthesis quality.
>
> > Similarly, for the public dataset comparison, the author did not select the SOTA models for the comparison. It would be great to see the comparison against them.
>
> In our current work, we included several representative and recent baselines that have demonstrated advanced performance in text-to-speech (TTS) tasks. These include VALL-E (2023), NaturalSpeech2 (2023), InstructTTS (2023), PromptTTS2 (2023), and VoiceCraft (2024). These methods encompass a diverse range of model architectures, including prompt-based TTS and codec model-based TTS approaches, and are among the most recognized in recent literature for achieving state-of-the-art results. If you have additional recommendations for models that you believe are critical for comparison, we would be delighted to consider them. Your suggestions would help us further enhance the comprehensiveness of our study.

---

### Official Review · Reviewer_SWYL · 2024-11-03

**Soundness:** 3
**Presentation:** 3
**Contribution:** 3
**Rating:** 6
**Confidence:** 4

**Summary:**

This paper introduces DNASpeech, a novel contextualized and situated text-to-speech (CS-TTS) dataset designed to enhance TTS performance by incorporating prompts from dialogues, narratives, and actions (DNA). DNASpeech provides rich multimodal prompts aligned with speech clips, filling a gap in current datasets that lack comprehensive contextual information for TTS tasks. The dataset includes 2,395 distinct characters, 4,452 scenes, and 22,975 dialogue utterances, along with over 18 hours of high-quality speech recordings. To validate DNASpeech, the authors propose a leaderboard and evaluation benchmarks featuring two subtasks: CS-TTS with narratives and CS-TTS with dialogues. The experimental results demonstrate the potential of DNASpeech to drive advancements in controllable and expressive TTS systems.

**Strengths:**

The integration of dialogues, narratives, and actions (DNA) as contextual prompts is an innovative addition to existing TTS datasets, providing richer and more varied situational context. The dataset is constructed using a detailed and well-validated annotation pipeline, with emphasis on quality control through denoising, ASR verification, and manual assessment. The alignment method that combines both coarse-grained and fine-grained techniques is robust and well-implemented. The dataset addresses a crucial need for more context-aware TTS systems and offers a structured evaluation through the established leaderboard. This contribution is likely to inspire further research in controllable TTS using diverse prompts. Additionally, the paper provides a thorough explanation of the dataset construction and the challenges involved, with the use of visual aids to depict the pipeline and dataset characteristics aiding in comprehension.

**Weaknesses:**

The experiments primarily focus on validating the dataset using specific subtasks (narratives and dialogues), but they could benefit from broader model diversity and more diverse metrics beyond MOS evaluations. Including results from a larger variety of baseline models would make the evaluation more comprehensive. While the paper claims that DNASpeech can generalize well for different TTS tasks, the experimental evidence supporting this claim is limited, and testing with a wider set of models and comparing performance on tasks beyond CS-TTS (e.g., emotional TTS) would strengthen this assertion. Additionally, the dataset's reliance on movie scripts might limit its applicability for general conversational TTS, as the movie-based context might not fully represent day-to-day conversational dynamics.

**Questions:**

Can the authors provide further insights into how well the dataset generalizes to tasks beyond CS-TTS, such as emotional speech synthesis or audiobook narration?

How do the authors address potential biases introduced by the use of movie scripts as the primary source for dataset construction?

Would additional metrics beyond MOS, such as word error rate (WER) or speaker similarity, provide more insights into the dataset's applicability?

---

> ### Author Response · Authors · 2024-11-24
> **Reply to Reviewer SWYL**
>
> **Thank you for taking the time and effort to review our submission. Below are our responses to the questions you raised.**
>
> - **Broader Model Diversity and Metrics:** While our experiments primarily focus on narratives and dialogues using Mean Opinion Score (MOS) evaluations, we acknowledge the value of exploring a broader set of baseline models and evaluation metrics. We have selected well-established TTS baselines spanning multiple categories (non-prompt, prompt-based, and codec model-based TTS systems) to evaluate DNASpeech. However, we agree that including additional models with different architectures and paradigms could enhance the evaluation's comprehensiveness. Incorporating alternative metrics, such as perceptual evaluations of speech quality (PESQ) or word error rate (WER), is an excellent suggestion for further validating model performance. We plan to include these aspects in follow-up work.
>
> - **Generalization Across Tasks Beyond CS-TTS:** We emphasize that DNASpeech introduces a novel CS-TTS framework that enables contextualized and situated speech synthesis, validated through comprehensive experiments on narratives and dialogues. However, we recognize the need for additional experimental evidence to substantiate claims of generalizability across diverse TTS tasks. Future work will explore applications like emotional TTS and expressive speech synthesis to highlight DNASpeech's adaptability. This will involve extending experiments to include tasks like emotional variability and speech styles, as well as utilizing models designed explicitly for such variations.
>
> - **Dataset Applicability and Contextual Limitations:** DNASpeech's reliance on movie scripts stems from the rich multimodal information and diverse scenarios they provide. While movie-based contexts may not capture all day-to-day conversational dynamics, our annotation pipeline's design ensures high-quality alignment between textual, contextual, and acoustic components. As mentioned in Appendix B, to address the concern of limited conversational applicability, we aim to diversify the dataset by incorporating data from real-world conversational settings and other domains in future expansions.

---

### Official Review · Reviewer_zq6R · 2024-11-06

**Soundness:** 3
**Presentation:** 2
**Contribution:** 3
**Rating:** 5
**Confidence:** 3

**Summary:**

This paper proposes a new TTS task and benchmark to produce contextualized and situated synthetic speech using dialogues, narratives and actions.

**Strengths:**

The new dataset would be useful to the TTS community and will be made available
The data creation methodology seems reasonable as does the design of the DNA Speech model.

**Weaknesses:**

1. The precise meaning of situated and contextualized is not very clear from my reading of the paper. Further, it is not very clear how actions in particular aid in situated and contextualized TTS.

2. From the data pipeline, it is not clear whether the obtained subtitles exactly match the speech, or are machine generated in some way. There seem to be many automated portions, for example, obtaining subtitles through OCR, getting Dialogues, Actions, Narratives and Characters from the original movie scripts, speech denoising etc. For all of these steps, there are no objective measures of quality reported, which casts doubt on the quality of data used.  Furthermore, the only quality evaluation used involves training a TTS models using DNASpeech and evaluating it.

3. The proposed ASR filtering based on Whisper could be potentially aggressive because the authors remove all non perfect matches. This means that the data obtained is selected based on Whisper's biases for movie transcription, which is not ideal.

**Questions:**

1. Three listeners is a small number for a TTS MOS test. Did the authors conduct a wider test ?

---

> ### Author Response · Authors · 2024-11-24
> **Reply to Reviewer zq6R (1/2)**
>
> **Sincerely thank you for your time and effort. Here are our reply to your concerns**
>
> > The precise meaning of situated and contextualized is not very clear from my reading of the paper.
>
> We define "situated" and "contextualized" as two complementary aspects of how speech generation can be enriched by surrounding information:
>
> - **Contextualized** refers to leveraging information that is directly related to the immediate conversational or narrative context of the speech. For instance, dialogue history or narrative descriptions provide cues about the tone, pace, and emotional state that should be reflected in synthesized speech.
>
> - **Situated** extends beyond the textual context to include environmental or situational factors influencing speech. This could include details about the scene (e.g., a "spooky forest") or the actions of the speaker (e.g., "whispering while hiding").
>
> > Further, it is not very clear how actions in particular aid in situated and contextualized TTS.
>
> Actions provide dynamic and real-time information about the speaker’s behavior and physical state during speech production. For example, a speaker described as "shouting angrily while pacing" provides cues not just about the emotional tone but also about the loudness, pitch, and urgency of the generated speech. Similarly, "speaking softly while looking down" can inform softer intonations and subdued vocal delivery. By integrating these action descriptors, the CS-TTS system synthesizes speech that aligns more closely with real-world scenarios, where vocal characteristics are inherently tied to physical gestures, emotional expressions, and situational context. This multifaceted approach ensures that the synthesized speech is not only contextually accurate but also vividly aligned with the surrounding environment and speaker's physical state.

---

> ### Author Response · Authors · 2024-11-24
> **Reply to Reviewer zq6R (2/2)**
>
> > From the data pipeline, it is not clear whether the obtained subtitles exactly match the speech, or are machine generated in some way. There seem to be many automated portions, for example, obtaining subtitles through OCR, getting Dialogues, Actions, Narratives and Characters from the original movie scripts, speech denoising etc. For all of these steps, there are no objective measures of quality reported, which casts doubt on the quality of data used. Furthermore, the only quality evaluation used involves training a TTS models using DNASpeech and evaluating it.
>
> To ensure scalability and generalizability in our data construction methods, we automate the process as much as possible. To maintain data quality, our pipeline incorporates multiple steps to accurately align subtitles, contextualized and situated information, and speeches. From the raw data (approximately 400,000 entries) to the final dataset (22,975 entries), only 6% of the data passed this rigorous screening process.
>
> Naturally, concerns regarding data quality are understandable. However, such concerns are unlikely to be alleviated unless a comprehensive manual verification of the dataset is conducted, which would be extremely time-consuming. Given the limited time for discussion, we sampled 200 entries and manually checked each for accuracy in matching. The results showed no errors in these 200 entries. The IDs of the sampled entries are as follows:
>
> *[138, 199, 345, 427, 437, 558, 579, 593, 721, 786, 1502, 1729, 1819, 2013, 2047, 2139, 2178, 2365, 2426, 2548, 2734, 2738, 2743, 2890, 3145, 3165, 3305, 3315, 3344, 3385, 3449, 3497, 3541, 3565, 3583, 3647, 3894, 3980, 4012, 4099, 4308, 4315, 4412, 4972, 5121, 5361, 5457, 5474, 5689, 5828, 5926, 5951, 6038, 6048, 6114, 6298, 6361, 6424, 6522, 6934, 6974, 7027, 7217, 7297, 7308, 7468, 7645, 7911, 7931, 8059, 8202, 8423, 8520, 8897, 9055, 9182, 9184, 9322, 9385, 9867, 9992, 10002, 10053, 10690, 10780, 10846, 10857, 10858, 10869, 11004, 11077, 11089, 11159, 11327, 11581, 11586, 11899, 11930, 12053, 12123, 12246, 12300, 12448, 12497, 12512, 12554, 12607, 12760, 12796, 13026, 13098, 13114, 13117, 13146, 13177, 13325, 13405, 13441, 13606, 13692, 13875, 14037, 14061, 14153, 14263, 14328, 14454, 14508, 14723, 14832, 14867, 15146, 15244, 15382, 15439, 15460, 15483, 15511, 15704, 15977, 16127, 16210, 16361, 16523, 16664, 16682, 16760, 16918, 16993, 17015, 17147, 17191, 17329, 17441, 17459, 17602, 17653, 17767, 17802, 17854, 17882, 18134, 18179, 18272, 18297, 18356, 18361, 18472, 18645, 18921, 19085, 19144, 19385, 19672, 19718, 19859, 19995, 20159, 20184, 20204, 20248, 20447, 20495, 21023, 21207, 21301, 21309, 21651, 21668, 21682, 21722, 21794, 22249, 22436, 22553, 22683, 22783, 22789, 22887, 22914]*
>
> Our dataset will be made publicly available, allowing this manual verification to be independently validated.
>
> > The proposed ASR filtering based on Whisper could be potentially aggressive because the authors remove all non perfect matches. This means that the data obtained is selected based on Whisper's biases for movie transcription, which is not ideal.
>
> Thank you for pointing out this issue. We believe the quality of data is more important than its quantity, which is why we have removed all non-perfect matches. We acknowledge that using Whisper for speech recognition is not a flawless solution. In our future work, we plan to integrate results from multiple ASR models to mitigate the bias inherent in a single model. If you have any better suggestions, please feel free to share with us.
>
> > Three listeners is a small number for a TTS MOS test. Did the authors conduct a wider test?
>
> Currently, due to practical constraints, we are unable to recruit more professional evaluators at this time. However, we would like to provide you with more detailed information about the three evaluators to address your concern about the number of participants:
>
> Three listeners participated in the evaluation process, each holding a master's degree and having completed prior training. After each round of testing, we calculate the Kendall's W coefficient for the scores provided by the three listeners. The results are accepted only when the Kendall's W coefficient $\geq 0.5$, ensuring consistency in the ratings.

---

> > ### Comment · Reviewer_zq6R · 2024-11-24
> >
> > ```
> > Our dataset will be made publicly available, allowing this manual verification to be independently validated.
> > ```
> > This does not appear like a reasonable solution to me, unfortunately. I am not sure how useful a dataset created by automatic labelling with insufficient human validation would be.
> >
> >
> > ```
> > Three listeners participated in the evaluation process, each holding a master's degree and having completed prior training. After each round of testing, we calculate the Kendall's W coefficient for the scores provided by the three listeners. The results are accepted only when the Kendall's W coefficient , ensuring consistency in the ratings.
> > ```
> >
> > Standard listening tests, to the best of my knowledge have between 5-10 annotators from my understanding. From that lens, 3 is definitely insufficient. I am not sure whether it is a good solution to filter ratings based on consistency across raters.
> >
> >
> > Based on the responses to my questions, and the two concerns I raised above, I am still in between acceptance and rejection for this paper, and will keep my score.

---

### Note · Authors · 2024-12-14

**Comment:**

Thank you to all the reviewers for your valuable feedback. We will continue to improve our work based on your suggestions.

**Withdrawal Confirmation:**

I have read and agree with the venue's withdrawal policy on behalf of myself and my co-authors.